# Investigating the added value of biomarkers compared with self-reported smoking in predicting future e-cigarette use: Evidence from a longitudinal UK cohort study

**Jasmine N. Khouja**[1,2,3]*, **Marcus R. Munafò**[1,3,4], **Caroline L. Relton**[1,2], **Amy E. Taylor**[2,4], **Suzanne H. Gage**[5], **Rebecca C. Richmond**[1,2]

**1** MRC Integrative Epidemiology Unit at the University of Bristol, Bristol, United Kingdom, **2** Bristol Medical School, Population Health Sciences, University of Bristol, Bristol, United Kingdom, **3** School of Psychological Science, University of Bristol, Bristol, United Kingdom, **4** NIHR Biomedical Research Centre at the University Hospitals Bristol NHS Foundation Trust and the University of Bristol, Bristol, United Kingdom, **5** Department of Psychological Sciences, University of Liverpool, Liverpool, United Kingdom

* Jasmine.Khouja@bristol.ac.uk

**Data Availability Statement:** Data cannot be shared publicly because the ALSPAC Ethics and Law Committee and the Local Research Ethics

## Abstract

Biomarkers can be used to assess smoking behaviour more accurately and objectively than self-report. This study assessed the association between cotinine (a biomarker of smoke exposure) and later e-cigarette use among a population who were unexposed to e-cigarettes in youth. Young people in the Avon Longitudinal Study of Parents and Children took part in the study. We observed associations between cotinine at 15 years (measured between 2006 and 2008 before the wide availability of e-cigarettes) and self-reported ever use of e-cigarettes at 22 (measured between 2014 and 2015 when e-cigarettes were widely available) using logistic regression. A range of potential confounders were adjusted for (age, sex, body mass index, alcohol use and passive smoke exposure). Additionally, we adjusted for the young people's self-reported smoking status/history to explore potential misreporting and measurement error. In a sample of N = 1,194 young people, cotinine levels consistent with active smoking at 15 years were associated with increased odds of e-cigarette ever use at 22 years (Odds Ratio [OR] = 7.24, 95% CI 3.29 to 15.93) even when self-reported active smoking status at age 16 (OR = 3.14, 95% CI 1.32 to 7.48) and latent classes of smoking behaviour from 14 to 16 (OR = 2.70, 95% CI 0.98 to 7.44) were included in the model. Cotinine levels consistent with smoking in adolescence were strongly associated with increased odds of later e-cigarette use, even after adjusting for reported smoking behaviour at age 16 and smoking transitions from 14 to 16.

## Introduction

There are an estimated 3.2 million e-cigarette users (also known as vapers) in the UK [1]. In 2015, just under 5% of 16 to 24 year olds were estimated to be current e-cigarette users in the

Committees have imposed restrictions on the data availability. However, the data used in this study can be made available on request to the ALSPAC Executive (alspac-exec@bristol.ac.uk). The ALSPAC data management plan describes in detail the policy regarding data sharing, which is through a system of managed open access. Full instructions for applying for data access can be found here: http://www.bristol.ac.uk/alspac/researchers/access/. The ALSPAC study website contains details of all the data that are available (http://www.bristol.ac.uk/alspac/researchers/our-data/). Access is subject to eligibility, the ALSPAC funder's terms and conditions and University of Bristol policies and procedures.

**Funding:** The UK Medical Research Council (https://mrc.ukri.org/) and Wellcome (https://wellcome.ac.uk/)(Grant ref: 102215/2/13/2) and the University of Bristol provide core support for ALSPAC. This publication is the work of the authors who will serve as guarantors for the contents of this paper. A comprehensive list of grants funding is available on the ALSPAC website (http://www.bristol.ac.uk/alspac/external/documents/grant-acknowledgements.pdf). The authors are supported by the UK Medical Research Council Integrative Epidemiology Unit at the University of Bristol (MC_UU_00011/7, MM_UU_00011/5). This work was also supported by CRUK (https://www.cancerresearchuk.org/) (grant numbers C18281/ A19169 and C57854/ A22171), Wellcome Trust and the UK Medical Research Council (grant number: 092731). RCR is a de Pass Vice Chancellor Research Fellow at the University of Bristol (http://www.bristol.ac.uk/vc-fellows/about/health/de-pass/). The funders had no role in study design, data collection and analysis, decision to publish, or preparation of the manuscript.

**Competing interests:** The authors have declared that no competing interests exist.

Health Survey for England [2]. Evidence suggests that e-cigarettes could be considerably less harmful than smoking [3] and that they can be effective in aiding smoking cessation [4]. Furthermore, frequent e-cigarette use among tobacco-naïve young people appears to be rare [5]. However, concerns have been raised about the use of e-cigarettes by non-smokers and given the potential harms of e-cigarette use (e.g., adverse pulmonary [6] and cardiovascular [7] effects), further investigation of the potential risk factors for e-cigarette use is warranted.

Previous research exploring associations between tobacco smoking and e-cigarette use among young people have generally relied on self-report data in cross-sectional surveys and longitudinal cohorts [8]. Self-reports of smoking behaviour have been shown to be less accurate than objective measures such as cotinine, which can increase detection of current smokers compared to self-report [9]. The high potential for measurement error when using self-reports of smoking means it is difficult to draw conclusions regarding the chronology of events i.e., that smoking preceded e-cigarette use or vice versa. Thus, there is limited objective evidence for whether smoking is associated with later e-cigarette use among young people or whether e-cigarette use is associated with later smoking among young people.

Cotinine is a direct metabolite of nicotine which can be used to assess recent smoke exposure (half-life of approximately 10–27 hours) [10]. It may also serve as an indicator for misreporting of smoking by providing biological evidence that smoking has occurred when individuals report that they have not smoked [11–13]. For example, one study reported that among self-reported non-smokers, 6% were misclassified and according to their cotinine levels had in fact recently smoked [12]. Therefore, cotinine could provide less biased evidence of association between smoking behaviour and later e-cigarette use.

To our knowledge, there have been no studies assessing the association of cotinine measurement in adolescence with e-cigarette use in early adulthood. We therefore examined the associations of cotinine with later ever e-cigarette use using biochemical verification of smoking status to further explore previous findings suggesting self-reported smoking is associated with later e-cigarette use. We also investigated any residual associations after accounting for self-reported smoking, in order to explore whether objectively assessed smoking is associated with e-cigarette use over and above self-reported smoking (which may be subject to misreporting).

## Materials and methods

### Sample

The study sample consisted of young people from the Avon Longitudinal Study of Parents and Children (ALSPAC) [14, 15]. A total of 15,454 pregnant women resident in Avon, UK with expected dates of delivery between 1st April 1991 to 31st December 1992 were recruited, resulting in 15,589 fetuses. Most of these women were recruited whilst pregnant; however, 913 women who had delivery dates within this time frame were recruited in later phases when the children were roughly 7 years and older. The phases of enrolment are described in more detail in the cohort profile update [16]. Of this total sample, 14,901 were alive at 1 year of age (see Fig 1). The study website contains details of all the data that is available through a fully searchable data dictionary: http://www.bris.ac.uk/alspac/researchers/data-access/data-dictionary/. Ethics approval for the study was obtained from the ALSPAC Ethics and Law Committee and the Local Research Ethics Committees. Informed consent for the use of data collected via questionnaires and clinics was obtained from participants following the recommendations of the ALSPAC Ethics and Law Committee at the time. Fig 1 shows the process from recruitment to the final sample selection.

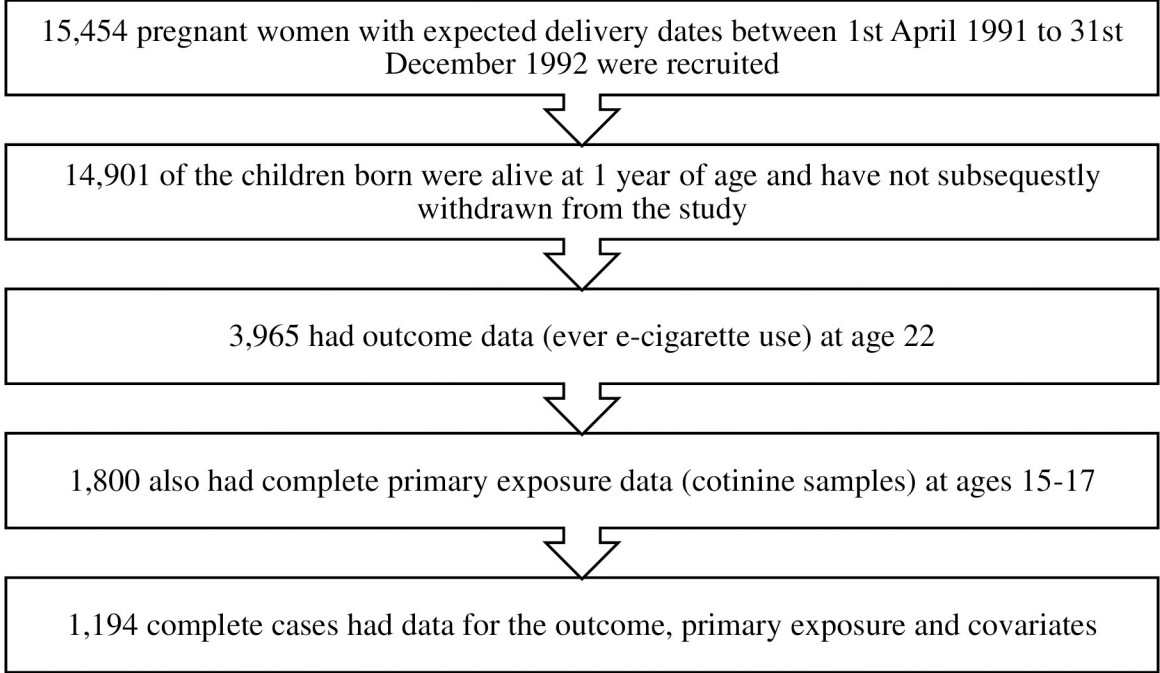

15,454 pregnant women with expected delivery dates between 1st April 1991 to 31st December 1992 were recruited

14,901 of the children born were alive at 1 year of age and have not subsequestly withdrawn from the study

3,965 had outcome data (ever e-cigarette use) at age 22

1,800 also had complete primary exposure data (cotinine samples) at ages 15-17

1,194 complete cases had data for the outcome, primary exposure and covariates

**Fig 1. Flow chart depicting the process of data inclusion in the analysis of the associations between cotinine samples at ages 15–17 and ever e-cigarette use at age 22.**

## Cotinine as a biomarker for predicting smoke exposure in adolescence

Cotinine is a commonly used biomarker for assessing nicotine consumption [17]. In this study, cotinine levels were measured from blood samples taken in a clinic assessment of the ALSPAC young people when they were 15 years old (between 2006 and 2008). Cotinine was assayed from ethylenediaminetetraacetic acid serum plasma samples which had been stored at −80˚C and allowed to thaw at room temperature before use. Cotinine was measured using the Cozart Cotinine Enzyme Immunoassay (Concateno UK, Abingdon) serum kit (M155B1). All samples, calibrators, and controls were brought to room temperature before use and were run in duplicate. Where required, samples were diluted using cotinine-free serum (fetal calf serum). Absorbance was measured spectrophotometrically at a wavelength of 450nm.

A presence of 10 ng/ml (nanograms per millilitre of serum) or more of cotinine in the blood stream is indicative of being an active smoker [18]. Levels below this but above 0 ng/ml would indicate some exposure to tobacco smoke and levels of 0 ng/ml would indicate no exposure. The young adults were categorised into three groups (0 ng/ml, no smoke exposure; between 0 and 10 ng/ml, passive smoke exposure; 10+ ng/ml, active smoking). Cotinine variables were grouped due to the non-linear association previously found between cotinine and smoke exposure [11, 19].

## Self-reported smoking in adolescence and early adulthood

Smoking status was assessed in adolescence (16 years) and in early adulthood (22 years). At 16 and 22 years, young people self-reported whether they had ever smoked a cigarette (yes/no) and the number of cigarettes they had ever smoked (less than 5, 5–19, 20–49, 100 plus). For analysis, the number of cigarettes ever smoked was recoded to include a 0 cigarettes category based on their response to ever smoking ("No" = 0). The frequency at which the young person smoked at age 16 was also measured (did not smoke, smoked once ever, used to smoke but did not at time of measure, smoked less than once a week, smoked 1–6 cigarettes a week, smoked

6+ cigarettes a week, smoked daily). This variable was also recoded into a binary variable for active (weekly/daily smoking) vs not active (all other responses) smoking. At 22 years, young people additionally self-reported whether they had smoked in the past 30 days (yes/no).

Due to the limitations of using secondary data we were unable to acquire measures of self-reported exposure at the same time as measures of cotinine. Self-reported measures were collected via questionnaire an average of 15 months after cotinine, which was measured at a clinic session. Time between the measures ranged between 0 and 29 months apart. To account for the potential variability in smoking behaviour over time in youth, Longitudinal Latent Class Analysis (LLCA) was also performed using *Mplus v8* [20] to derive smoking classes based on a four-category ordinal variable with categories 'none', 'less than weekly', 'weekly' and 'daily' smoking from three questionnaire time points (14, 15 and 16 years), as has been described previously [21]. A 4-class model was selected for the repeated measures of smoking frequency, comprising smoking behaviour patterns that we refer to as never-smokers (84%), experimenters (6%), late onset regular smokers (9%) and early onset regular smokers (2%). These classes are consistent with those derived previously by Heron et al. [21], where non-smokers reported very little or no smoking, experimenters smoked infrequently (monthly), late-onset regular smokers were individuals who began smoking by age 14 and were mostly daily smokers by age 16 and early-onset regular smokers were mostly daily smokers by age 14. Given the very high entropy of the 4-class model, individuals were assigned to the class for which they had the highest probability of class membership (modal class assignment) and the latent classes were used as an observed categorical exposure in further analyses.

## E-cigarette use in adulthood

E-cigarette use was determined by self-report questionnaire during 2014–2015 when the young people were 22 years old. Study data at 22 years were collected and managed using REDCap electronic data capture tools hosted at the University of Bristol [22]. Participants were given a yes/no option to the question: "have you ever used/smoked/vaped an electronic cigarette?" Additional information was collected regarding current (past 30-day) use of e-cigarettes but these data were not used as the main outcome due to sample size constraints.

## Covariates

A variety of covariates were included that could impact on the association between biomarkers of smoking and later e-cigarette use; sex (male/female), age at the time of questionnaire (quartiles younger to older), parental education (at 32 weeks of gestation; Degree, A-level, O-level, less than O-level; defined in S1 File), body mass index (BMI at 15 years old), parental social class (at 18 weeks of gestation; manual, non-manual), passive smoke exposure (maternal smoking at 12 years old) and alcohol use (whether the young person had drunk a whole alcoholic beverage in the past 30 days at 15 years old). Self-reported ever cannabis use (at 15 years) was included as a covariate in supplementary analyses. Self-reported smoke exposure (whether the young person had ever smoked, the number of cigarettes smoked, whether they were active smokers at 16 years, and latent classes of smoking transitions between the ages of 14, 15 and 16 [latent classes: never smokers, experimenters, late onset regular smokers, early onset regular smokers; within class probabilities are shown in S1–S4 Figs]) was also included in further models.

## Statistical analyses

Analyses were completed in Stata SE 15.1 (Stata Corp LP, College Station, TX USA). Differences in sociodemographic characteristics between the young people who had never used an e-cigarette versus those who had ever used an e-cigarette by 22 years were assessed using a chi

square test for categorical variables and t-test for continuous variables. Associations between cotinine at age 15 and ever e-cigarette use at age 22 years were assessed in a series of logistic regressions. Cases with missing data (exposure, outcome or covariate data) were not included in the analyses.

Analysis of cotinine data was performed on both categorised (main text) and continuous (supplementary) data. Emphasis has been placed on the analysis of the categorical data for two reasons: 1) the continuous cotinine data were positively skewed and attempts at transforming the data did not sufficiently address the issue; and 2) non-linear relationships between cotinine and smoke exposure may mean that continuous models might not be informative [11, 19].

Three models were used to analyse the associations between cotinine and later e-cigarette use before adjusting for self-reported smoking; the basic model (Model 1) adjusted for sex and age at cotinine sample collection; Model 2 additionally adjusted for BMI, socio-economic position (social class, maternal and paternal education) and alcohol use; Model 3 additionally adjusted for passive smoke exposure (maternal smoking) at 12 years. A further four models were used to analyse the association while adjusting for self-reported smoking. These further models were as Model 3 and additionally adjusted for ever smoking at age 16 (Model 4a), number of cigarettes smoked in their lifetime at 16 (Model 4b), active smoking (daily/weekly) at age 16 (Model 4c), or smoking transitions from age 14 to 16 (Model 4d). On average, the time between the measure of cotinine at 15 years and the self-reported measure of smoking status at 16 years was 15 months, with a range of 0 to 29 months. Given the short half-life of cotinine and the potential variability in smoking behaviour over time in youth, Models 4a-4c also adjusted for the time (months) between cotinine measure and self-report at 16. Any associations observed in Models 1–3 were expected to attenuate to the null in Models 4a-4d in the absence of measurement error in the self-report.

Model fit was assessed using likelihood ratio tests in which the models including and excluding the exposure (cotinine) were compared. Substantial differences between the two models indicate that the model including cotinine was more predictive of the outcome (i.e., the model was a good fit).

A sensitivity analysis was conducted to explore whether time between measures of cotinine and self-reported smoking behaviour may have influenced the results. We explored whether restricting the analysis to those with a time gap of 18 months or less between the smoking measures would affect the interpretation of the results.

## Results

### Characteristics of participants

At 22 years, the question regarding ever e-cigarette use was completed by 3,965 participants, 955 (24%) of whom responded that they had previously used/smoked/vaped an e-cigarette. Of those who had ever used an e-cigarette, 111 (12%) stated that they were currently using one. Table 1 contains the participants' sociodemographic data stratified by their use of e-cigarettes (ever/never) at 22 years. There was strong evidence for differences between groups in terms of sex, BMI, alcohol use and smoking behaviour (at 16 and 22 years). Among never users of e-cigarettes at 22 years, 37% had ever smoked at 16 and 48% had smoked at 22 years (self-reported). For ever e-cigarette users at 22 years, 74% had ever smoked at age 16 and 95% had ever smoked at age 22 (self-reported). There was also weak evidence that maternal education differed between groups.

### Cotinine levels and e-cigarette ever use at 22 years

The main results for the associations between cotinine and e-cigarette ever use can be found in Table 2. Levels of cotinine indicative of passive smoke exposure at 15 years were not clearly

**Table 1. Sociodemographic profile of the sample population.**

| Variable | Never used an e-cigarette at 22 years (n = 3,010) | | Had ever used an e-cigarette by 22 years (n = 955) | | |
|---|---|---|---|---|---|
| | n/Mean | %/SD | n/Mean | SD/% | p-value |
| Male | 1000 | 33% | 367 | 38% | .003 |
| Age in months at cotinine measure (15 years)* | 185 | 4 | 185 | 4 | .42 |
| BMI at 15 years* | 21 | 3 | 22 | 4 | .001 |
| Social class (non-manual) | 2,314 | 86% | 713 | 84% | .24 |
| Maternal education | | | | | .024 |
| Degree or above | 584 | 21% | 147 | 17% | |
| A level | 769 | 28% | 248 | 29% | |
| O level | 929 | 34% | 300 | 35% | |
| Less than O level | 459 | 17% | 170 | 20% | |
| Paternal education | | | | | .12 |
| Degree or above | 767 | 29% | 205 | 24% | |
| A level | 770 | 29% | 250 | 30% | |
| O level | 554 | 21% | 187 | 22% | |
| Less than O level | 599 | 22% | 202 | 24% | |
| Alcohol use in the past 30 days at 15 | 1,211 | 57% | 476 | 76% | < .001 |
| Ever smoked at age 16 | 851 | 37% | 481 | 74% | < .001 |
| Number of cigarettes smoked by age 16 | | | | | < .001 |
| 0 | 1,472 | 64% | 167 | 26% | |
| Less than 5 | 403 | 17% | 123 | 19% | |
| 5–19 | 154 | 7% | 82 | 13% | |
| 20–49 | 95 | 4% | 56 | 9% | |
| 50–99 | 66 | 3% | 49 | 8% | |
| 100 or more | 123 | 5% | 170 | 26% | |
| Active (daily/weekly) smoking at age 16 | 123 | 5% | 188 | 29% | < .001 |
| Ever smoked at age 22 | 1,443 | 48% | 908 | 95% | < .001 |
| Current smoker at age 22 | 412 | 14% | 588 | 62% | < .001 |
| Number of cigarettes smoked by age 22 | | | | | < .001 |
| 0 | 1,534 | 52% | 46 | 5% | |
| Less than 5 | 301 | 10% | 40 | 4% | |
| 5–19 | 310 | 11% | 64 | 7% | |
| 20–49 | 218 | 7% | 57 | 6% | |
| 50–99 | 144 | 5% | 59 | 6% | |
| 100 or more | 444 | 15% | 670 | 72% | |

Samples relate to the young people unless otherwise stated. *P*-values from chi square test for categorical variables and *t*-test for continuous variables.

*Mean and standard deviation (SD) are presented for these variables. All other variables display the number of participants (n) and percentage of the population.

associated with ever e-cigarette use at 22 years ($p > .28$). Levels of cotinine indicative of active smoking at 15 years were associated with a 10-fold increase in the odds of having ever used e-cigarettes at 22, compared with those with no smoke exposure (basic adjusted OR 10.47, 95% CI 4.88 to 22.46, $p < .001$). This positive association remained following additional adjustment, although was weakened when adjusting for self-reported smoking behaviour at age 16 (ever smoking, OR 5.00, 95% CI 2.25 to 11.14, $p < .001$; number of cigarettes smoked, OR 2.35, 95% CI 0.98 to 5.62, $p = .055$; active [weekly/daily] smoking, OR 3.15, 95% CI 1.32 to 7.48, $p = .010$; and smoking transitions, OR 2.70, 95% CI 0.98, 7.44, $p = .054$). Including cotinine in the model improved the model fit for Models 1, 2, 3, 4a and 4c ($p < .001$), but not 4b

**Table 2. Associations of cotinine at 15 years and ever use of e-cigarettes at 22 years (N = 1,194).**

| Model | Cotinine | | | | | |
|---|---|---|---|---|---|---|
| | Passive (n = 797) | | | Active (n = 36) | | |
| | OR | 95% CI | *p*-value | OR | 95% CI | *p*-value |
| 1 | 1.19 | 0.87, 1.63 | .28 | 10.47 | 4.88, 22.46 | < .001 |
| 2 | 1.13 | 0.81, 1.56 | .47 | 8.06 | 3.69, 17.61 | < .001 |
| 3 | 1.10 | 0.80, 1.52 | .56 | 7.24 | 3.29, 15.93 | < .001 |
| 4a | 1.15 | 0.82, 1.61 | .42 | 5.00 | 2.25, 11.14 | < .001 |
| 4b | 1.07 | 0.76, 1.51 | .71 | 2.35 | 0.98, 5.62 | .055 |
| 4c | 1.06 | 0.76, 1.48 | .71 | 3.14 | 1.32, 7.48 | .010 |
| 4d | 1.05 | 0.75, 1.46 | .79 | 2.70 | 0.98, 7.44 | .054 |

Reference group = no exposure; OR = odds ratio; 95% CI = 95% confidence interval. Cotinine was treated as a categorical variable in these analyses. Passive exposure is defined as levels exceeding 1 ng/ml in blood samples (up to 9 ng/ml). Active exposure is defined as cotinine levels exceeding 10 ng/ml in blood samples. The basic model (model 1) was adjusted for age and sex. Model 2 was additionally adjusted for socioeconomic status, BMI and alcohol. Model 3 was additionally adjusted for passive smoke exposure (maternal smoking at 12 years). Models 4a-4c were as model 3 and additionally adjusted for self-reported measures of smoking and the difference in age between the self-report and cotinine measures. Model 4a adjusted for ever smoking at age 16. Model 4b alternatively adjusted for number of cigarettes smoked by age 16. Model 4c alternatively adjusted for active smoking (daily/weekly) at age 16. Model 4d was as Model 3 and adjusted for classes of smoking transitions; early onset regular smokers, late onset regular smokers, never smokers and experimenters categorised using data from 14 to 16.

(likelihood ratio = 3.87, *p* = .14) or 4d (likelihood ratio = 3.91, *p* = .14). The evidence of an association was much weaker in the analysis of the continuous data (S1 Table). Including cannabis as a covariate further attenuated the association but some residual association was still seen (S2 Table).

## Self-reported smoking and potential misreporting

Although strong associations were seen between self-reported smoking behaviour and smoking behaviour determined by cotinine (Table 3), discrepancies indicate that misreporting may have occurred (Table 4). Only 56% of those who had cotinine levels indicative of being an active smoker self-reported daily smoking. 19% of those indicated to be active smokers by their cotinine levels reported that they were current non-smokers and 25% reported their smoking behaviour as weekly or less. Conversely, only 36% of those who self-reported being daily smokers also had cotinine levels indicating active smoking. Furthermore, e-cigarette use at 22 years was more strongly associated with cotinine levels indicating active smoking (OR adjusted for age and sex = 10.47, 95% CI 4.88 to 22.46, *p* < .001), than with self-reported active daily/weekly smoking at 15 years (OR adjusted for age and sex = 7.77, 95% CI 5.09 to 11.85, *p* < .001), albeit with overlapping confidence intervals (Table 5).

## Transitions in smoking and ever e-cigarette use at 22 years

The association among the different latent classes of transitions in smoking between 14 and 16 years and e-cigarette use at 22 years (adjusting for all model 3 covariates, including cotinine) are shown in Table 6. Experimenters and late onset regular smokers had increased odds of ever using an e-cigarette at 22 years. There was no clear difference in odds of ever e-cigarette use between early onset regular smokers and never smokers.

## Sensitivity analyses

We next evaluated whether the discrepancies between self-reported smoking and cotinine levels were due to the fact that cotinine measures were taken between 0 and 29 months after the

**Table 3. Associations of cotinine and smoking behaviours in a series of logistic regressions.**

| Model | 1 | | | 2 | | | 3 | | |
|---|---|---|---|---|---|---|---|---|---|
| Smoking behaviour | OR | 95% CI | p | OR | 95% CI | p | OR | 95% CI | p |
| Ever smoked a cigarette at age 16 (n = 1794) | | | | | | | | | |
| No cotinine exposure (reference) | 1.00 | - | - | 1.00 | - | - | 1.00 | - | - |
| Passive cotinine exposure | 1.06 | 0.86, 1.30 | .60 | 0.95 | 0.76, 1.19 | .68 | 0.93 | 0.74, 1.17 | .53 |
| Active cotinine smoking | 94.97 | 13.07, 689.89 | < .001 | 70.29 | 9.55, 517.40 | < .001 | 64.31 | 8.71, 475.06 | < .001 |
| Number of cigarettes smoked by age 16 (n = 1786) | | | | | | | | | |
| No cotinine exposure (reference) | 1.00 | - | - | 1.00 | - | - | 1.00 | - | - |
| Passive cotinine exposure | 1.13 | 0.93, 1.38 | .21 | 1.01 | 0.82, 1.24 | .91 | 0.99 | 0.81, 1.22 | .96 |
| Active cotinine smoking | 70.68 | 36.62, 136.43 | < .001 | 53.46 | 27.20, 105.06 | < .001 | 49.01 | 24.84, 96.73 | < .001 |
| Active (daily/weekly) smoking at age 16 (n = 1829) | | | | | | | | | |
| No cotinine exposure (reference) | 1.00 | - | - | 1.00 | - | - | 1.00 | - | - |
| Passive cotinine exposure | 1.48 | 0.98, 2.23 | .064 | 1.27 | 0.84, 1.94 | .26 | 1.24 | 0.81, 1.89 | .33 |
| Active cotinine smoking | 43.21 | 23.12, 80.77 | < .001 | 32.34 | 16.94, 61.72 | < .001 | 28.70 | 14.93, 55.18 | < .001 |
| Ever smoked a cigarette at age 22 (n = 1307)* | | | | | | | | | |
| No cotinine exposure (reference) | 1.00 | - | - | 1.00 | - | - | 1.00 | - | - |
| Passive cotinine exposure | 1.12 | 0.88, 1.43 | .35 | 1.07 | 0.83, 1.38 | .62 | 1.05 | 0.81, 1.36 | .72 |
| Number of cigarettes smoked by age 22 (n = 1345) | | | | | | | | | |
| No cotinine exposure (reference) | 1.00 | - | - | 1.00 | - | - | 1.00 | - | - |
| Passive cotinine exposure | 1.17 | 0.95, 1.46 | .14 | 1.14 | 0.91, 1.42 | .24 | 1.12 | 0.90, 1.40 | .32 |
| Active cotinine smoking | 30.50 | 11.89, 78.24 | < .001 | 22.21 | 8.55, 57.68 | < .001 | 20.99 | 8.06, 54.67 | < .001 |
| Current (past 30 day) smoker at age 22 (n = 1345) | | | | | | | | | |
| No cotinine exposure (reference) | 1.00 | - | - | 1.00 | - | - | 1.00 | - | - |
| Passive cotinine exposure | 1.23 | 0.92, 1.64 | .17 | 1.19 | 0.88, 1.61 | 0.247 | 1.18 | 0.87, 1.59 | .29 |
| Active cotinine smoking | 8.90 | 4.61, 17.20 | < .001 | 7.20 | 3.64, 14.25 | < .001 | 6.87 | 3.46, 13.66 | < .001 |

OR = odds ratio; 95% CI = 95% confidence interval. Cotinine was treated as a categorical variable in these analyses so ORs reflect the odds of each smoking behaviour for each cotinine level. Passive exposure is defined as levels exceeding 1 ng/ml up to 9 ng/ml in blood samples. Active smoking is defined as cotinine levels exceeding 10 ng/ml in blood samples. The basic model (model 1) was adjusted for age and sex. Model 2 was additionally adjusted for socioeconomic status, BMI and alcohol. Model 3 was additionally adjusted for passive smoke exposure (maternal smoking at 12 years).

*Cotinine levels indicating active smoking at 15 are not included in the table as they perfectly predicted whether young people had ever smoked at age 22.

16 year questionnaire on smoking behaviour in which smoking status was self-reported, and therefore due to changing smoking habits in that time rather than misreporting. Although the evidence of a residual association after adjusting for self-reported smoking status was weaker, there was still some evidence of an association in the sensitivity analysis (S3 Table), indicating that the residual association is not due to the time gap between the measures. The restriction

**Table 4. Self-reported frequency of smoking at 16 and smoking status indicated by cotinine (ng/ml) level at 15 years (n = 1,194).**

| Cotinine indication | Frequency of smoking | | | |
|---|---|---|---|---|
| | Never or not current | Current less than daily | Current daily | Total |
| No exposure | 311 (26%) | 44 (4%) | 6 (<1%) | 361 |
| Some exposure | 661 (55%) | 106 (9%) | 30 (3%) | 797 |
| Indicated smoker | 7 (<1%) | 9 (<1%) | 20 (2%) | 36 |

Some exposure is defined as levels exceeding 1 ng/ml up to 9 ng/ml in blood samples. Indicated smoker is defined as cotinine levels exceeding 10 ng/ml in blood samples.

**Table 5. Associations of active smoking (determined by cotinine levels or self-report) at 15 years and ever use of e-cigarettes at 22 years (N = 1,194).**

| Model | Active Smoking | | | | | |
|---|---|---|---|---|---|---|
| | Self-Report* | | | Cotinine | | |
| | OR | 95% CI | *p*-value | OR | 95% CI | *p*-value |
| 1 | 7.77 | 5.09, 11.85 | < .001 | 10.47 | 4.88, 22.46 | < .001 |
| 2 | 6.99 | 4.50, 10.86 | < .001 | 8.06 | 3.69, 17.61 | < .001 |
| 3 | 6.34 | 4.26, 10.34 | < .001 | 7.24 | 3.29, 15.93 | < .001 |

Self-report reference group = self-reported not current smoking; cotinine reference group = no exposure indicated by cotinine levels; OR = odds ratio; 95% CI = 95% confidence interval. Cotinine was treated as a categorical variable in these analyses. Active exposure is defined as cotinine levels exceeding 10 ng/ml in blood samples; no cotinine exposure is defined as cotinine levels of 0 ng/ml in blood samples. The basic model (model 1) was adjusted for age and sex. Model 2 was additionally adjusted for socioeconomic status, BMI and alcohol. Model 3 was additionally adjusted for passive smoke exposure (maternal smoking at 12 years).

reduced the sample size and therefore the power to detect an association, so seeing a consistent residual association (albeit with weaker statistical evidence) is supportive of our interpretation of the results.

## Discussion

To our knowledge, this is the first longitudinal study of the association between cotinine in adolescence and later e-cigarette use in early adulthood. Cotinine levels that are indicative of active smoking at age 15 are strongly associated with increased odds of having ever used an e-cigarette at age 22. Although the strength of evidence of this association was weakened after adjustment for self-reported smoking behaviour at age 16 (assessed based on ever smoking status, number of cigarettes smoked, active smoking and latent classes of smoking), some evidence of association remained, indicating that some measurement error had occurred which could potentially be due to misreporting.

The evidence provides further support (using biochemical verification) to research findings suggesting that smoking is associated with later e-cigarette use. There was little evidence for an association between cotinine levels indicative of passive exposure with later e-cigarette use. This reflects previous research suggesting e-cigarette use is more common in smokers than non-smokers [23, 24]. As causality cannot be inferred from associations, we cannot determine whether smoking in adolescence causes future e-cigarette use. However, if there is a true causal association the results provide some indication of the direction of causality due to the timing of the data collection. When the data were collected (2006–2008), e-cigarettes were newly available and in 2013 (over 5 years later when the first evidence of use among young people

**Table 6. Associations of latent classes of transitions in smoking between 14 and 16 years and ever use of e-cigarettes at 22 years (N = 1,194).**

| Latent classes of smoking | OR | 95% CI | *p*-value |
|---|---|---|---|
| Never smokers (ref) | 1 | - | - |
| Experimenters | 4.13 | 2.29, 7.47 | < .001 |
| Late onset regular smokers | 6.22 | 3.13, 12.36 | < .001 |
| Early onset regular smokers | 0.92 | 0.11, 7.72 | 0.94 |

OR = odds ratio; 95% CI = 95% confidence interval. Latent classes of smoking transitions (early onset regular smokers, late onset regular smokers, never smokers and experimenters) were categorised using data from questionnaires at 14, 15 and 16 years. The model adjusted for age, sex, socioeconomic status, BMI and alcohol, passive smoke exposure (maternal smoking at 12 years) and cotinine.

began to emerge) only 4.6% of 11–18 year olds had ever used e-cigarettes in Great Britain [25]. Therefore, it is very likely that these biomarker profiles reflect smoking behaviour in adolescence without confounding from e-cigarette use, given that e-cigarettes were not widely used among youths when the biomarker levels were measured.

If self-reported smoking and biomarker indications of smoking were both accurate and no misreporting had occurred, the association between cotinine levels and later e-cigarette use should have been fully attenuated when adjusting for self-reported smoking. The residual association indicates that smoking might be misreported in adolescence. Particularly, under-reporting of smoking may be an issue when assessing self-reported smoke exposure; nearly 20% of the young people in this study who were identified as having cotinine levels indicating active smoking reported not actively smoking. Previous research has also found that misreporting of smoking in young people is common [26]. There are many possible explanations for misreporting when gathering self-report data on smoking behaviour from young people, such as social pressures, fear of punishment or recall bias [27]. Therefore, research using self-report data to explore the gateway hypothesis (that e-cigarettes act as a gateway to smoking) may not be reliable as young people reporting to be 'non-smokers' who use e-cigarettes may have previously smoked cigarettes.

Measurement error could also explain the residual association between smoking and later e-cigarette use after adjusting for self-reported smoking. While there was an improvement in the prediction of e-cigarette use when cotinine was included in the models with ever and active smoking at age 16, there was less improvement when cotinine was added to models including more detailed assessment of smoking intensity and duration. This suggests that in future studies, in the absence of biomarker verification when investigating the association between e-cigarette use and smoking, more detailed self-report measures (e.g. transitions) should be used rather than cruder measures (e.g. ever use) to avoid measurement error.

In addition, 36% of individuals identified as non-smokers by cotinine levels self-reported being smokers. This discrepancy could have arisen from measurement error in cotinine (given its short half-life [10], the measure may have been taken too early/late) or the cut-off used to determine whether the young people's cotinine levels were indicative of active smoking or not may also have been too high. Despite 10 ng/mL frequently being used as the cut-off with high sensitivity and specificity [28], some studies have used cut-offs from 1 ng/mL [29] up to 25 ng/mL [30]. These discrepancies could be explained by changes in the young person's smoking status between the measures of self-reported smoking status and cotinine as they were measured between 0 and 29 months apart, with an average of 15 months. However, we adjusted for the time between measures in sensitivity analyses that indicated that the remaining association after adjusting for smoking status is unlikely to be due to the time gap between the measures. We also accounted for variability in smoking behaviour over time by assessing the latent classes, which captured transition in smoking between age 14 and 16.

Focussing on this model (Model 4d), we found that early onset regular smoking was not clearly associated with later e-cigarette use but experimentation and late onset regular smoking was, even when cotinine was taken into account. This finding suggests an additional element of smoking behaviour in predicting future e-cigarette use, which is not captured by cotinine measures. In particular, results indicate that those individuals who have smoked for a longer period of time (early onset smokers) are less likely to try e-cigarettes than experimenters and late-onset smokers, and are no more likely to try e-cigarettes in the future than never smokers. This highlights a difference between certain groups of the population in likelihood to engage in e-cigarette use. Further research should explore this as those who start smoking earlier may be more addicted to nicotine and may need more encouragement to quit whether that be by using e-cigarettes or other methods.

## Implications of findings and limitations

Longitudinal studies exploring the gateway hypothesis often rely on self-reported smoking history to exclude those who have already been exposed to smoking at baseline (prior to e-cigarette use) but it is also important to explore whether differences in biomarker levels that indicate smoking are evident. Individuals who claim to have used e-cigarettes before smoking cigarettes may be inaccurately reporting information and this misreporting could be captured using biomarker verification. However, now that e-cigarettes are more popular and have been shown to be a common correlate of cigarette smoking [31], replicating these results and avoiding problems of reverse causation may prove difficult as e-cigarettes can contain nicotine and therefore their use will increase cotinine levels.

The association between cotinine and e-cigarette use remained after including a range of potential confounders in the model. However other sources of nicotine may be confounding this association. Cannabis users in the UK often smoke cannabis with tobacco and may or may not report this as smoking. When cannabis use was included in the model the association between smoking and later e-cigarette use was somewhat attenuated. We did not include cannabis in the main analysis due to the potential for cannabis acting as an indicator of misreporting rather than a confounder. The role of cannabis in the relationship between smoking and e-cigarette use should be further explored. Another source of nicotine exposure which may not have been fully captured in the model is passive smoking. Although we adjusted for maternal smoking, the measure was assessed 3 years prior to the exposure and there are numerous other sources through which the young person could have been passively exposed to smoke.

Ideally, we would have further explored the potential bias resulting from misreporting of smoke exposure in studies exploring whether e-cigarettes act as a gateway to smoking by restricting our analyses to those who claimed they were never smokers but were regular e-cigarette users at 22 years. We were unable to do this due to the limited number of young people in this study who fit those criteria. Only 1% of the sample had tried e-cigarettes but claimed to have never smoked at 22 years and less than 1% of the sample were current e-cigarette users and claimed to have never smoked a cigarette. The low number of non-smoking regular e-cigarette users is reflected in recent statistics from the England-based Smoking Toolkit Study [32] and the literature exploring the gateway hypothesis (e.g. Primack and colleagues [33] identified only 16 non-smoking ever e-cigarette users among 694 participants).

## Conclusions

In summary, cotinine levels consistent with active smoking in adolescence are associated with later e-cigarette use even after adjusting for some measures of self-reported smoking behaviour. This could have implications for studies assessing the gateway hypothesis that rely on self-report measures of smoking behaviour. Future studies investigating prediction of e-cigarette use should investigate use of more detailed self-report and objective measures.

## Supporting information

**S1 Fig. Probability of being a daily, weekly, long-term weekly or never smoker in latent class 1.**
(PDF)

**S2 Fig. Probability of being a daily, weekly, long-term weekly or never smoker in latent class 2.**
(PDF)

**S3 Fig. Probability of being a daily, weekly, long-term weekly or never smoker in latent class 3.**
(PDF)

**S4 Fig. Probability of being a daily, weekly, long-term weekly or never smoker in latent class 4.**
(PDF)

**S1 Table. Associations of cotinine (continuous) at 15 years and ever use of e-cigarettes at 22 years (N = 1,194).** Reference group = no exposure; OR = odds ratio; 95% CI = 95% confidence interval. Cotinine was treated as a continuous variable in these analyses (ng/ml in blood samples). The basic model (model 1) was adjusted for age and sex. Model 2 was additionally adjusted for socioeconomic status, BMI and alcohol. Model 3 was additionally adjusted for passive smoke exposure (maternal smoking at 12 years). Models 4a-4c were as model 3 and additionally adjusted for self-reported measures of smoking and the difference in age between the selfreport and cotinine measures. Model 4a adjusted for ever smoking at age 16. Model 4b alternatively adjusted for number of cigarettes smoked by age 16. Model 4c alternatively adjusted for active smoking (daily/weekly) at age 16. Model 4d was as Model 3 and adjusted for classes of smoking transitions; early onset regular smokers, late onset regular smokers, never smokers and experimenters categorised using data from 14 to 16.
(PDF)

**S2 Table. Associations of cotinine at 15 years and ever use of e-cigarettes at 22 years (N = 1,194) including cannabis use as a covariate.** Reference group = no exposure; OR = odds ratio; 95% CI = 95% confidence interval. Cotinine was treated as a categorical variable in these analyses. Active exposure is defined as cotinine levels exceeding 10 ng/ml in blood samples. The basic model (model 1) was adjusted for age and sex. Model 2 was additionally adjusted for socioeconomic status, BMI, alcohol and cannabis use. Models 4a-4c were as model 3 and additionally adjusted for various self-reported measures of smoking and the difference in age between the cotinine measure and the self-report. Model 4a adjusted for ever smoking at age 16. Model 4b alternatively adjusted for number of cigarettes smoked by age 16. Model 4c alternatively adjusted for active smoking (daily/weekly) at age 16. Model 4d was as model 3 and adjusted for classes of smoking transitions; early onset regular smokers, late onset regular smokers, never smokers and experimenters categorised using data from 14 to 16.
(PDF)

**S3 Table. Associations of cotinine at 15 years and ever use of e-cigarettes at 22 years restricting to those with age differences below 18 months (n = 870).** Reference group = no exposure; OR = odds ratio; 95% CI = 95% confidence interval. Cotinine was treated as a categorical variable in these analyses. Active exposure is defined as cotinine levels exceeding 10 ng/ml in blood samples. The basic model (model 1) was adjusted for age and sex. Model 2 was additionally adjusted for socioeconomic status, BMI and alcohol. Model 3 was additionally adjusted for passive smoke exposure (maternal smoking at 12 years). Models 4a-4c were as model 3 and additionally adjusted for self-reported measures of smoking and the difference in age between the self-report and cotinine measures. Model 4a adjusted for ever smoking at age 16. Model 4b alternatively adjusted for number of cigarettes smoked by age 16. Model 4c alternatively adjusted for active smoking (daily/weekly) at age 16. Model 4d was as model 3 and adjusted for classes of smoking transitions; early onset regular smokers, late onset regular smokers, never smokers and experimenters categorised using data from 14 to 16.
(PDF)

**S1 File. Parental education.**
(PDF)

## Acknowledgments

We are extremely grateful to all the families who took part in this study, the midwives for their help in recruiting them, and the whole ALSPAC team, which includes interviewers, computer and laboratory technicians, clerical workers, research scientists, volunteers, managers, receptionists and nurses.

## Author Contributions

**Conceptualization:** Jasmine N. Khouja, Marcus R. Munafò, Caroline L. Relton, Amy E. Taylor, Suzanne H. Gage, Rebecca C. Richmond.

**Data curation:** Jasmine N. Khouja, Rebecca C. Richmond.

**Formal analysis:** Jasmine N. Khouja.

**Funding acquisition:** Amy E. Taylor.

**Investigation:** Jasmine N. Khouja.

**Methodology:** Jasmine N. Khouja, Rebecca C. Richmond.

**Supervision:** Rebecca C. Richmond.

**Writing – original draft:** Jasmine N. Khouja.

**Writing – review & editing:** Jasmine N. Khouja, Marcus R. Munafò, Caroline L. Relton, Amy E. Taylor, Suzanne H. Gage, Rebecca C. Richmond.

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
