## [Decision Letter · Decision Letter 0]

23 Apr 2020

PONE-D-20-06112

Investigating the added value of biomarkers compared with self-reported smoking in predicting future e-cigarette use: Evidence from a longitudinal UK cohort study

PLOS ONE

Dear Miss Khouja,

Thank you for submitting your manuscript to PLOS ONE. After careful consideration, we feel that it has merit but does not fully meet PLOS ONE’s publication criteria as it currently stands. Therefore, we invite you to submit a revised version of the manuscript that addresses the points raised during the review process.

We would appreciate receiving your revised manuscript by Jun 07 2020 11:59PM. To enhance the reproducibility of your results, we recommend that if applicable you deposit your laboratory protocols in protocols.io, where a protocol can be assigned its own identifier (DOI) such that it can be cited independently in the future. For instructions see: http://journals.plos.org/plosone/s/submission-guidelines#loc-laboratory-protocols

We look forward to receiving your revised manuscript.

Kind regards,

Antonio Palazón-Bru, PhD

Academic Editor

PLOS ONE

Journal Requirements:

1. We note that you have indicated that data from this study are available upon request. PLOS only allows data to be available upon request if there are legal or ethical restrictions on sharing data publicly. For information on unacceptable data access restrictions, please see http://journals.plos.org/plosone/s/data-availability#loc-unacceptable-data-access-restrictions.

Reviewers' comments:

Reviewer's Responses to Questions

**Comments to the Author**

1. Is the manuscript technically sound, and do the data support the conclusions?

Reviewer #1: Yes

Reviewer #2: Yes

Reviewer #3: Partly

2. Has the statistical analysis been performed appropriately and rigorously? 

Reviewer #1: Yes

Reviewer #2: Yes

Reviewer #3: Yes

3. Have the authors made all data underlying the findings in their manuscript fully available?

Reviewer #1: Yes

Reviewer #2: Yes

Reviewer #3: Yes

4. Is the manuscript presented in an intelligible fashion and written in standard English?

Reviewer #1: Yes

Reviewer #2: Yes

Reviewer #3: Yes

5. Review Comments to the Author

Reviewer #1: In this article by Khouja et al, the authors used data from a cohort of young adults (ALPSAC) who were assessed for tobacco exposure using cotinine at age 15 and e-cigarette use using self-report at age 22. The authors found that cotinine levels were associated with higher risk of ever e-cigarette use, which was attenuated after adjustment for cigarette smoking patterns. Those who were labeled as “experimenters” and late-onset regular smokers had increased odds of ever using e-cigarettes at age 22.

This study is important in that it demonstrates that there may be a temporal relationship between early cigarette use in adolescent and subsequent e-cigarette use in early adulthood. This reviewer suspects that the results are likely still confounded given the marked attenuation in measures of association after subsequent adjustment for cigarette use patterns. However no prior study had such granular data about tobacco exposure patterns at multiple points in time to correlate traditional cigarette use with future e-cigarette use. The authors are to be commended for this effort though this reviewer considers these results to be hypothesis generating given possible residual confounding and the use of multiple time points in this study. This certainly justifies the need for future studies with more frequent and timely assessment longitudinal changes in cigarette and e-cigarette use.

While the long-term safety of e-cigarettes remains unclear, one cannot conclusively label e-cigarettes as being “safer” than cigarettes in the absence of long-term data. Of particular interest is pulmonary and cardiovascular complications associated with e-cigarette use. There may also be psychosocial comorbidities associated with e-cigarette use, There is therefore a concern that warrants the identification of risk factors associated with e-cigarette use.

The authors also found discrepancies between objectively measured cigarette use (cotinine) and subjective measured use (self-report) whereby more than half of participants with cotinine levels indicative of active cigarette use indicated that they were current daily smokers. These results are certainly important and point to the limitations of self-reported data with regards to tobacco use be it cigarettes or e-cigarettes.

The authors of the present study showed a higher risk of e-cigarette use with prior cigarette use. Do the authors also have follow up data at age 22 to determine the association between e-cigarette and cigarette use?

Did the authors also examine the association between cigarette and subsequent current e-cigarette use (rather than ever)? These data would be interesting to see. Along similar lines, is there data on vaping patterns among current e-cigarette users?

Reviewer #2: Dear authors, I have reviewed your manuscript and I only have two minor suggestions

1.L.256: the table S3 is very inclusive and informative, therefore it would be helpful for the reader if it was included in the results section in the main body of the manuscript (not as a supplementary material) .

2. L.381-387: please describe your study's limitations clearly as so, under a Limitations subtitle.

Reviewer #3: The main claim of the paper is that biomarkers such as cotinine are more accurate than self-reported smoking in predicting future e-cigarette use. In my opinion the best way for the authors to assess that would be by performing a comparison between the smoking status in the age of 16 based on cotinine levels and the smoking status in the age of 16 based on self-report using chi-square in the population of e-cigarette vapers in the age of 22.

In the 1st paragraph of the “Introduction” of their paper the authors state that: “Evidence suggests that e-cigarettes are considerably less harmful than smoking …........ frequent e-cigarette use among tobacco-naïve young people is rare”. Existing evidence are conflicting as far as: 1) The harmfullness of e-cigarette compared to tobacco cigarette, 2) The effectiveness of e-cigarette in aiding smoking cessation, and 3) The frequency of e-cigarette use among tobacco-naïve young people. Thus, in my opinion, based on the existing evidence, the authors should have had a neutral attitude concerning e-cigarette role on those aspects rather than a possitive one.

In the 2nd paragraph of the “Discussion” of their paper the authors state that: “Furthermore, this evidence does not support the theory …... the relationship is more complex”. In my opinion such a conclusion cannot be supported by the results of this research. Such a conclusion could only come of by a research that would investigate the long-term smoking status of young people who started vaping without being tobacco cigarette smokers. Thus, in my opinion, this sentence should be drawn out of this paper.

In the 6th paragraph of the “Discussion” of their paper (lines 346 – 355) the authors report that “early onset regular smoking was not clearly associated with later e-cigarette use but experimentation and late onset regular smoking was”. This was a finding in a period in which e-cigarette was not widely available. Nowadays, when e-cigarette is widely available, it is possible that a proportion of “late onset regular smokers” and/or “experimenters” might start vaping instead of smoking and turn to smoking or dual use later. In my opinion this should be discussed in that paragraph.

6. PLOS authors have the option to publish the peer review history of their article (what does this mean?). If published, this will include your full peer review and any attached files.

Reviewer #1: Yes: Mahmoud Al Rifai

Reviewer #2: No

Reviewer #3: No

---

## [Author Response · Author response to Decision Letter 0]

10 Jun 2020

Dear Dr. Palazón-Bru,

We would like to thank the reviewers for their time and efforts reviewing our manuscript. We have addressed their comments and provided details of edits made to the manuscript in light of the suggestions made.

Reviewer #1

1. This study is important in that it demonstrates that there may be a temporal relationship between early cigarette use in adolescent and subsequent e-cigarette use in early adulthood. This reviewer suspects that the results are likely still confounded given the marked attenuation in measures of association after subsequent adjustment for cigarette use patterns. However, no prior study had such granular data about tobacco exposure patterns at multiple points in time to correlate traditional cigarette use with future e-cigarette use. The authors are to be commended for this effort though this reviewer considers these results to be hypothesis generating given possible residual confounding and the use of multiple time points in this study. This certainly justifies the need for future studies with more frequent and timely assessment longitudinal changes in cigarette and e-cigarette use. While the long-term safety of e-cigarettes remains unclear, one cannot conclusively label e-cigarettes as being “safer” than cigarettes in the absence of long-term data. Of particular interest is pulmonary and cardiovascular complications associated with e-cigarette use. There may also be psychosocial comorbidities associated with e-cigarette use. There is therefore a concern that warrants the identification of risk factors associated with e-cigarette use. The authors also found discrepancies between objectively measured cigarette use (cotinine) and subjective measured use (self-report) whereby more than half of participants with cotinine levels indicative of active cigarette use indicated that they were current daily smokers. These results are certainly important and point to the limitations of self-reported data with regards to tobacco use be it cigarettes or e-cigarettes.

RESPONSE: We thank the reviewer for their positive comments about the importance and novelty of the study. We agree that these findings can be viewed as exploratory and hypothesis-generating, with the need for more research to investigate temporal relationships between smoking and e-cigarette use. However, we would like to emphasise the value of the longitudinal approach taken in our study, with the ability to rule out reverse causality. We also agree that discrepancies between cotinine and self-report identified in this analysis are of interest, particularly among those who report not smoking but who have cotinine levels consistent with being an active smoker (with the caveat of the differences in time of assessment). We appreciate the concern regarding the statements made around the safety of e-cigarettes, and have made the description more neutral since health risks have not been evaluated in this study. Page 4, line 49: “Evidence suggests that e-cigarettes could be considerably less harmful than smoking [3] and that they can be effective in aiding smoking cessation [4]. Furthermore, frequent e-cigarette use among tobacco-naïve young people appears to be rare [5]. However, concerns have been raised about the use of e-cigarettes by non-smokers and given the potential harms of e-cigarette use (e.g., adverse pulmonary [6] and cardiovascular [7] effects), further investigation of the potential risk factors for e-cigarette use is warranted.”

2. The authors of the present study showed a higher risk of e-cigarette use with prior cigarette use. Do the authors also have follow up data at age 22 to determine the association between e-cigarette and cigarette use? 

RESPONSE: In table 1, we show that ever e-cigarette users are more likely to have ever smoked than never users by 22 years (95% and 48% respectively; p<.001) and ever users were more likely to currently smoke than never users at 22 years (62% and 14% respectively; p<.001). We also provide the data at 16 years (Table 1) for comparison. Unadjusted odds ratios can be calculated with the data provided, but the main aim of the study was to assess cotinine levels in relation to future e-cigarette use. As described in our discussion, cotinine assessment at age 22 would not be able to distinguish cigarette from e-cigarette use, which was of importance to our study. 

3. Did the authors also examine the association between cigarette and subsequent current e-cigarette use (rather than ever)? These data would be interesting to see. Along similar lines, is there data on vaping patterns among current e-cigarette users? 

RESPONSE: There are too few current e-cigarette users in ALSPAC (n = 111) who also have cotinine measures at 15 (n = 41) and also have data for all covariates (n= 28) to conduct any meaningful analyses on current e-cigarette users at 22 years. Limited data on patterns of use were collected at 22 years meaning we are unable to assess patterns of use.

Reviewer #2

4. L.256: the table S3 is very inclusive and informative, therefore it would be helpful for the reader if it was included in the results section in the main body of the manuscript (not as a supplementary material). 

RESPONSE: We thank the reviewer for their suggestions, we have moved Table S3 to the main body of the manuscript (now Table 3).

5. L.381-387: please describe your study's limitations clearly as so, under a Limitations subtitle.

RESPONSE: We have added a limitations subtitle (page 22, line 413).

Reviewer #3

6. The main claim of the paper is that biomarkers such as cotinine are more accurate than self-reported smoking in predicting future e-cigarette use. In my opinion the best way for the authors to assess that would be by performing a comparison between the smoking status in the age of 16 based on cotinine levels and the smoking status in the age of 16 based on self-report using chi-square in the population of e-cigarette vapers in the age of 22. 

RESPONSE: We thank the reviewer for their comments and agree that a comparison between the prediction of self-report versus cotinine would be useful so we have added the following (page 14, line 283): “Furthermore, e-cigarette use at 22 years was more strongly associated with cotinine levels indicating active smoking (OR adjusted for age and sex = 10.47, 95% CI 4.88 to 22.46, p < .001), than with self-reported active daily/weekly smoking at 15 years (OR adjusted for age and sex = 7.77, 95% CI 5.09 to 11.85, p < .001), albeit with overlapping confidence intervals (Table 5).”

7. In the 1st paragraph of the “Introduction” of their paper the authors state that: “Evidence suggests that e-cigarettes are considerably less harmful than smoking …........ frequent e-cigarette use among tobacco-naïve young people is rare”. Existing evidence are conflicting as far as: 1) The harmfulness of e-cigarette compared to tobacco cigarette, 2) The effectiveness of e-cigarette in aiding smoking cessation, and 3) The frequency of e-cigarette use among tobacco-naïve young people. Thus, in my opinion, based on the existing evidence, the authors should have had a neutral attitude concerning e-cigarette role on those aspects rather than a positive one. 

RESPONSE: In line with the reviewer’s advice and the advice of Reviewer 1, we have amended this paragraph (see first response to Reviewer 1).

8. In the 2nd paragraph of the “Discussion” of their paper the authors state that: “Furthermore, this evidence does not support the theory …... the relationship is more complex”. In my opinion such a conclusion cannot be supported by the results of this research. Such a conclusion could only come of by a research that would investigate the long-term smoking status of young people who started vaping without being tobacco cigarette smokers. Thus, in my opinion, this sentence should be drawn out of this paper. 

RESPONSE: As suggested, we have removed this sentence from the manuscript.

9. In the 6th paragraph of the “Discussion” of their paper (lines 346 – 355) the authors report that “early onset regular smoking was not clearly associated with later e-cigarette use but experimentation and late onset regular smoking was”. This was a finding in a period in which e-cigarette was not widely available. Nowadays, when e-cigarette is widely available, it is possible that a proportion of “late onset regular smokers” and/or “experimenters” might start vaping instead of smoking and turn to smoking or dual use later. In my opinion this should be discussed in that paragraph. 

RESPONSE: This is an interesting hypothesis, but we feel this would be too speculative to include based on our results. However, on page 22 (line 408) we have re-emphasised that finding those who smoke infrequently or started smoking later in life are more likely to try e-cigarettes compared with early onset smokers “highlights a difference between certain groups of the population in likelihood to engage in e-cigarette use.”

---

## [Decision Letter · Decision Letter 1]

19 Jun 2020

Investigating the added value of biomarkers compared with self-reported smoking in predicting future e-cigarette use: Evidence from a longitudinal UK cohort study

PONE-D-20-06112R1

Dear Dr. Khouja,

We’re pleased to inform you that your manuscript has been judged scientifically suitable for publication and will be formally accepted for publication once it meets all outstanding technical requirements.

Kind regards,

Antonio Palazón-Bru, PhD

Academic Editor

PLOS ONE

Additional Editor Comments (optional):

Reviewers' comments:

Reviewer's Responses to Questions

**Comments to the Author**

1. If the authors have adequately addressed your comments raised in a previous round of review and you feel that this manuscript is now acceptable for publication, you may indicate that here to bypass the “Comments to the Author” section, enter your conflict of interest statement in the “Confidential to Editor” section, and submit your "Accept" recommendation.

Reviewer #1: All comments have been addressed

Reviewer #3: All comments have been addressed

2. Is the manuscript technically sound, and do the data support the conclusions?

Reviewer #1: Yes

Reviewer #3: Yes

3. Has the statistical analysis been performed appropriately and rigorously? 

Reviewer #1: Yes

Reviewer #3: Yes

4. Have the authors made all data underlying the findings in their manuscript fully available?

Reviewer #1: Yes

Reviewer #3: Yes

5. Is the manuscript presented in an intelligible fashion and written in standard English?

Reviewer #1: Yes

Reviewer #3: Yes

6. Review Comments to the Author

Reviewer #1: The authors have satisfactorily answered my inquiries. I do not have additional comments and I think the paper is now suitable for publication.

Reviewer #3: (No Response)

7. PLOS authors have the option to publish the peer review history of their article (what does this mean?). If published, this will include your full peer review and any attached files.

Reviewer #1: Yes: Mahmoud Al Rifai

Reviewer #3: No

---

## [Editor Report · Acceptance letter]

26 Jun 2020

PONE-D-20-06112R1 

Investigating the added value of biomarkers compared with self-reported smoking in predicting future e-cigarette use: Evidence from a longitudinal UK cohort study 

Dear Dr. Khouja:

I'm pleased to inform you that your manuscript has been deemed suitable for publication in PLOS ONE. Congratulations! Your manuscript is now with our production department. 

Kind regards, 

on behalf of

Dr. Antonio Palazón-Bru 

Academic Editor

PLOS ONE